# Australian Youth Resilience and Help-Seeking during COVID-19: A Cross-Sectional Study

**DOI:** 10.3390/bs13020121

**Published:** 2023-02-01

**Authors:** Christine Grove, Alexandra Marinucci, Ilaria Montagni

**Affiliations:** 1Fulbright Association, Canberra, NSW 2601, Australia; 2Faculty of Education, Monash University, Clayton, VIC 3800, Australia; 3Bordeaux Population Health Research Center UMRS1219, University of Bordeaux, Inserm, F-33000 Bordeaux, France

**Keywords:** resilience, youth, COVID-19, help-seeking, mental health

## Abstract

The COVID-19 pandemic has seriously impacted youth mental health. Their resilience, defined as the ability to respond to adversity, has also been impaired. Help-seeking refers to the activity of addressing oneself to others when facing trouble. The objective of this study was to understand the levels of youth resilience and help-seeking during COVID-19 in 2021. Data were collected online from 181 Australian adolescents aged 12–17 years. The General Help-Seeking Questionnaire, the Actual Help-Seeking Questionnaire, and the Resilience Scale were used. Mean and frequency analysis and independent samples t-tests were performed. The Pearson correlation coefficient was calculated. Resilience was in the low range (mean = 66.56, SD 15.74) and associated with no help-seeking. For a personal problem and suicidal ideation, participants were most likely to contact a mental health professional, with means of 4.97 (SD 1.75) and 4.88 (SD 1.97), respectively. The majority did not seek help (*n* = 47) for challenges with anxiety or depression. This study corroborates previous findings on limited help-seeking in youth because of self-reliance and low confidence in others. Resilience decreased during COVID-19 in parallel with help-seeking. Strategies aiming to increase resilience and help-seeking, such as school-based programs, are needed given their decrease in Australian youths due to the COVID-19 pandemic.

## 1. Introduction

Mental illness affects young people [1], and the rate of poor mental health is constantly increasing worldwide [2,3]. One in seven youths aged 10 to 19 years old experiences a mental illness, with depression and anxiety among the leading causes of illness and disability [4]. COVID-19 and reduced social interaction have impacted youth identity development, social relationships, and connectedness [3,5,6]. In Australia, government-imposed restrictions were enforced to prevent movement and interaction within and between states in 2020 and 2021. Restrictions varied by state, with Victoria experiencing the most frequent and most severe limitations [7]. During this time, schools were closed, with students moving to online learning. This period has been found to contribute to mental distress [8,9]. For Australian youths, the prevalence of anxiety and depression symptoms increased during restrictions compared with before COVID-19 [10]. Young people aged 16 to 25 years appeared to be more vulnerable to the effects of the pandemic on their mental health than older people [11,12]. Regardless of the presumed vulnerability, even well-adjusted youths (e.g., those with healthy family relationships, established social relationships, and a lack of existing mental illness) also experienced an increase in mental health problems during COVID-19 [13].

Resilience is the ability to overcome adversities and achieve better outcomes despite experiences of stressors and trauma [14,15,16]. This is influenced by context, culture, age, gender, and individual life circumstances [17].

Child and youth developments can affect resilience [18]. The combination of supportive relationships, social competence, positive experiences, problem solving skills, critical consciousness, autonomy, and a sense of purpose are the foundations of developing resilience and can be considered protective factors [19]. According to Resiliency Theory [20], factors that increase resilience include high self-efficacy and self-esteem, as well as resources such as opportunities for learning and support [21]. The protective factor model of resilience, guided by Resiliency Theory, posits that protective and risk factors interact. Therefore, by increasing protective factors, this can reduce potential negative outcomes despite adverse life events [21,22,23]. Resilience pertains to how one manages during times of adversity, and young people tend to cope better when adverse events are balanced with resilience factors, such as problem-solving skills and accessible support [24]. Resilience has been shown to be strongly associated with mental health in youth [25]. Youth who respond well to difficult situations usually have a biological resistance to adversity and strong protective factors such as supportive relationships with trusted adults and their community [26]. Prior to COVID-19, resilience was found to be in the moderate range (M = 70.48) on the Resilience Scale (RS-14) in an international sample of adolescents aged 13–17 years [27]. However, during COVID-19, resilience was found to be significantly impacted in international samples [28].

Help-seeking refers to active behaviours to address a problem by seeking support from others [29,30,31]. This might be informal, for example friends and family, or formal, such as a professional. Rickwood et al. [31] conceptualised help-seeking as a process of moving from personal to interpersonal behaviour, with formal support seeking occurring in four steps: awareness and appraisal of mental health difficulties; articulation of difficulties to others; available sources of help; and willingness to seek help and disclose difficulties. During COVID-19, given the rising rate of psychological distress in Australian youth [32], increasing their help-seeking behaviour for mental health challenges was paramount. A prospective cohort study found that challenges with depression and anxiety increased after the first wave of COVID-19 for adolescent Australians; however, there was no improvement in help-seeking behaviour [33]. Therefore, improving the availability and awareness of mental health programs, services, and support is vital, as research continues into risk and protective factors to address services for this age group [33]. Before COVID-19, the Longitudinal Study of Australian Children found that 1 in 10 young people sought help from a mental health professional, and their willingness to seek informal help decreased from the age of 10 to 15 years [34]. Young people have reported that barriers to seeking help include low mental health and help source knowledge, mental health stigma, and preference for self-reliance [35,36,37]. Prevention and early intervention are critical to improving youth mental health and are associated with better outcomes into adulthood [38]. Early intervention can prevent developing mental illness as the child grows into adolescence and adulthood [39]. Intervention strategies include cognitive–behavioural relaxation, social skills training, behavioural and social support, and mindfulness [40]. These have been recognized as opportunities to alter the trajectory of mental [39]. Young people are less likely to seek help than adults [41]. Common barriers for youth actually seeking help include negative attitudes towards mental illness; family distrust towards health professionals, including previous negative experiences; low mental health literacy to identifying when help is needed, how, and by who; and adolescents’ need of autonomy and a concern of potential confidentiality breaches [42]. Given these barriers to actually seeking help and their vulnerable developmental stage [43], adolescents are most in need of help for mental health [44].

The aim of this study was to understand how young people sought help for mental health during COVID-19 in 2021 and their levels of resilience. This was guided by three research questions: (1) Where did young people seek help during COVID-19?; (2) Where would young people intend to seek help during COVID-19?; and (3) How resilient were young people during COVID-19?

## 2. Materials and Methods

### 2.1. Participants

The researchers contacted schools, and used snowballing and advertising to recruit participants. Young people in mainstream secondary schools in Australia were invited to participate in this study directly through their school or through social media advertisements. Their data were gathered with consent from their guardian and their assent. Inclusion criteria for this study were those aged 12 to 17 years, attending a mainstream secondary school, and could understand written English. Data were collected online from July to October 2021 using Qualtrics (Qualtrics, 2021; Provo, UT, USA), with 201 responses recorded. Participants with more than 10% missing data were removed (*n* = 20); 181 participants were included in the study.

Various ethnicities were included in the sample. Australian participants comprised the majority of the sample (*n* = 102, 56.4%), with the remaining participants stating their ethnicity as Asian (*n* = 32, 17.7%), Mixed (2 or more ethnicities; *n* = 25, 13.8%), African (*n* = 5, 2.8%), European (*n* = 3, 1.7%), British (*n* = 2, 1.1%), Middle Eastern (*n* = 2, 1.1%), American (*n* = 1, 0.6%), and Latin American (*n* = 1, 0.6%). Eight participants did not state their ethnicity. The average age of participants was 13.76 years (SD = 1.28, range = 12–17) and the average grade was 8.25 (SD = 1.32, range = 7–12). The ratio of male (*n* = 85) to female (*n* = 87) participants was balanced, with five participants describing their gender as non-binary, two participants identifying as transgender, and two participants responding ‘other’. Most participants resided in Victoria (*n* = 153), with the remaining residing in New South Wales (*n* = 4), Queensland (*n* = 3), Tasmania (*n* = 2), Western Australia (*n* = 2), and Australian Capital Territory (*n* = 1). Sixteen participants did not state where they lived. Participants attended non-government/private schools (*n* = 146), government schools (*n* = 19), or Catholic schools (*n* = 16).

### 2.2. Measures

#### 2.2.1. Help-Seeking

Help-seeking was measured through two components: intention to seek for help and actual help-seeking. Two scales were used to measure help-seeking to generate a clearer understanding of how young people seek help and to compare intention with actual behaviour. Intention to seek help was measured using the General Help-Seeking Questionnaire (GHSQ) [31,45]. This has been validated for use with Australian adolescents with a Cronbach’s alpha of 0.85 and test–retest reliability of 0.92 [45]. The GHSQ did not demonstrate clear evidence of adequate measurement model fit or internal consistency in the present sample [46].

For the purpose of this study, the GHSQ followed a matrix format in response to 2 items (“If you were having a personal or emotional problem, how likely is it that you would seek help from the following people?” and “If you were experiencing suicidal thoughts, how likely is it that you would seek help from the following people?”) with sources of help rated on a 7-point Likert scale (1 = Extremely Unlikely, 2 = Very Unlikely, 3 = Unlikely, 4 = Somewhat Likely, 5 = Likely, 6 = Very Likely, 7 = Extremely Likely). The sources of help in this study were friend, partner, parent, relative, mental health professional, helpline, doctor, religious leader, would not seek help, and other, which provided an option to enter other sources of help not included in the questionnaire.

Actual help-seeking was measured using the Actual Help-Seeking Questionnaire (AHSQ) [45,47]. The AHSQ asks participants to state who they have turned to for advice or help in the past 2 weeks for a personal or emotional problem using a multi-choice response format. The choices were friend, parent, relative, partner, mental health professional, helpline, doctor, teacher, not sought help, or other, where the participant could provide other sources of help they sought in an open question response box. Participants provided information of the type of problem they went to help for each selected source.

#### 2.2.2. Resilience

Resilience was measured using the 14-item Resilience Scale (RS-14) [48], which has been validated for use with American adolescents with a Cronbach’s alpha of 0.91 [49]. Items are measured along a 7-point Likert scale with statements rated from Strongly Disagree to Strongly Agree (e.g., “I can usually find something to laugh about.”). Lower scores indicate lower levels of resilience, with a range of scores from 14 to 98 [50].

### 2.3. Data Analysis

Analysis was conducted through mean and frequency analysis using SPSS version 28 [51] with a significance level at >0.05. To determine significant differences for resilience (RS-14) between participants who had sought help/not sought help, sought help from parents/not sought help from parents, and sought help from friends/not sought help from friends (AHSQ), an independent samples t-test was conducted. Correlation analysis was conducted using the Pearson correlation coefficient to determine associations between resilience and help-seeking intentions.

## 3. Results

### 3.1. Help-Seeking

Table 1 displays the responses for the sample to the GHSQ, used to measure the participants’ intention to seek help from the listed sources. Participants were most likely to seek help from a mental health professional, parent, relative, or friend for a personal problem. For suicidal ideation, participants were most likely to seek help from a mental health professional, parent, or friend. These sources fell between “somewhat likely” and “likely” to seek help, with no sources rated as “extremely likely” on average in the sample. Participants were least likely to seek help from a religious leader, to not seek help, or “other”. For those who selected “other” for a personal problem, participants reported they would seek help from God (*n* = 3), a sport coach (*n* = 2), a friend’s parent (*n* = 1), social media (*n* = 1), the internet (*n* = 1), a teacher (*n* = 14), a random stranger (*n* = 3), or school (*n* = 1). For suicidal ideation, participants reported “other” as a football coach (*n* = 1), God (*n* = 2), a teacher (*n* = 11), neighbours (*n* = 1), a random stranger (*n* = 3), or school (*n* = 1). The Cronbach alpha for the GHSQ was 0.846.

Participants’ actual help-seeking in the past two weeks was measured using the AHSQ; Figure 1 displays the frequency of responses for each source. Participants reported they most frequently sought help from a friend or parent. However, many participants (*n* = 47) did not seek help.

Participants reported what they had sought help for in each source. Seeking help from a partner, friend, parent, or relative were reported for problems related to anxiety, stress, depression, relationship problems, and problems at school. Participants reported that they had sought help from a mental health professional or doctor for problems related to anxiety and depression. A teacher was sought help from for anxiety and stress, and a helpline was sought help from for depression, suicidal ideation, homelessness, and family problems.

Table 2 details each problem, indicating how many participants had sought help for each source.

One participant reported that they had sought help from their partner for self-esteem related to their attention deficit hyperactivity disorder (ADHD), “feeling worthless and annoying because I cannot always control my ADHD symptoms”. Participants who reported general wellbeing reported they had sought help for “how my day has been…if I am fine or not”. Relationship problems referred to friendship difficulties, arguments with friends or siblings, and loss of love in an intimate relationship. Changing identity referred to a participant who had sought help on changing their name. Behaviour was reported by one participant who had sought help for “how to act in within some situations”. Stress was reported as related to social media, school, friendships, and family. Participants reported they had sought help surrounding anxiety related to lockdown and returning to school after lockdown. Some participants reported that they had sought help for “everything”.

For participants who had reported “other”, they had sought help for “being homeless”, “emotions”, “my secrets, stress in life and [evil] desires”, and “unstable thoughts”.

### 3.2. Resilience

Resilience, measured by the RS-14, was found to be in the low range (mean = 66.56, SD = 15.74, range = 22–98, *n* = 174). Figure 2 displays the spread of scores across the sample. For help-seeking intentions for a personal problem, resilience was significantly correlated with seeking help from an intimate partner (*r* (170) = 0.21, *p* = 0.01), parent (*r* (169) = 0.40, *p* < 0.001), relative (*r* (172) = 0.44, *p* < 0.001), mental health professional (*r* (172) = 0.20, *p =* 0.01), and religious leader (*r* (172) = 0.22, *p =* 0.00). Resilience was significantly correlated with not seeking help for a personal problem (*r* (172) = −0.35, *p* < 0.001). For help-seeking intentions for suicidal ideation, resilience was significantly correlated with seeking help from an intimate partner (*r* (171) = 0.32, *p* < 0.01), a friend (*r* (120) = 0.38, *p* < 0.01), a parent (*r* (172) = 0.46, *p* < 0.01), a relative (*r* (172) = 0.45, *p* < 0.01), a mental health professional (*r* (172) = 0.34, *p* < 0.01), a helpline (*r* (172) = 0.25, *p* < 0.01), and doctor (*r* (172) = 0.30, *p* < 0.01), and a religious leader (*r* (171) = 0.30, *p* < 0.01). Resilience was significantly correlated with not seeking help for suicidal ideation (*r* (170) = −0.37, *p* < 0.01). The Cronbach’s alpha for the RS was 0.906.

No significant differences were found on resilience for participants who had or had not sought help from parents or friends. No significant differences were found for participants who had or had not reported to not seek help on resilience.

## 4. Discussion

This study aimed to understand the relationship between help-seeking and resilience among Australian youth during COVID-19.

### 4.1. Actual Help-Seeking

Most young people reported that they had sought help in the past two weeks from a friend or parent. Approximately one-quarter of young people in the sample reported that they had not sought help. The most frequent reasons for seeking help from a friend or parent were for anxiety and depression. Young people also reported that they sought help from a friend or a parent regarding suicide, for themselves and others. Help from a mental health professional was sought by a smaller number of young people, and this often centred around severe mental health problems such as trauma, self-harm, and suicide.

Our results are consistent with previous studies finding that young people are still less likely to seek help from adults or professionals due to negative attitudes towards mental health and self-reliance for symptom management [24,31,41,45,47,52].

### 4.2. Intended Help-Seeking

Young people in this study reported that they were most likely to intend to seek help for a personal problem from a mental health professional, parent, relative, or friend. For suicidal ideation, young people reported that they were most likely to intend to seek help from a mental health professional, parent, or a friend. Although these sources of help were reported the most likely, no sources of help were rated as “very likely” or “extremely likely”. This may indicate the reluctance of young people to be confident in where they could seek help in times of crisis [37,53]. COVID-19 had a negative impact on the mental health of young people, with youths feeling uncertain and without access to usual support mechanisms [54]. The findings in this study are congruent with the Mission Australia Youth Survey report [54], which found that most young people reported that they would seek help from a friend or parent, with a lower proportion reporting that they would seek help from a doctor or health professional.

### 4.3. Resilience

The level of resilience of young people in this sample was low, and compared with previous studies [49], had decreased. Pritzker and Minter [49] found that youth aged 11 to 14 years old had a mean level of resilience of 76.79 (SD = 13.03, *n* = 1184), and youth aged 15 to 19 years old had a mean level of resilience of 77.42 (SD = 14.20, *n* = 1624) on the RS-14. An Australian study of 16- and 17-year-old youths found that resilience, as measured by the Connor–Davidson Resilience Scale [55], was at an ‘average’ level, with young people reporting that they ‘often’ viewed themselves as resilient [56]. Prior to COVID-19, it appears young people were able to cope with everyday stressors and persist when faced with adversity [56]. Coping and resilience are associated with mental health; Pedrini et al. [3] found that lower resilience was a predictor of worsened mental health during COVID-19. More than one-half of young people have reported that they reduced stress by doing something relaxing, spending time online or playing games or watching TV/movies during COVID-19 [54]. A small amount of young people reported that they would seek help from a mental health professional to reduce stress. However, youth also reported that they would consume alcohol or other drugs to reduce stress. Young people report that a major barrier to seeking help and using positive coping strategies is low mental health literacy [35,57]. Low mental health literacy is a significant barrier to youth actually seeking help, and particularly being able to recognize a change in mental health, how to seek help, and who to seek help from [42]. Thus, strategies aimed to increase mental health literacy, resilience, and coping are needed given the decrease in Australian youth’s current level of resilience found in this study [36,58,59].

### 4.4. Strengths and Future Research

There are some strengths of this study. Firstly, a diverse sample was collected across Australia, with participants from different cultural backgrounds. This aligns with the multicultural landscape of Australia and allows perspectives from a range of individuals. Secondly, this may be one of the first studies to explore help-seeking intentions and actual help-seeking among Australian youths during COVID-19 and their levels of resilience. Resilience and help-seeking are related [60]; therefore, this study provides a rationale for supporting youth wellbeing through interventions aimed to increase resilience and actual help-seeking behaviours of mental health challenges arise. Future research may seek to identify ways of addressing the barriers to youth actually seeking professional support when needed.

The mean age of the participants in the study was 13.76 years. It is likely that the supports needed by 13-year-olds are different to that of older adolescents. Further research may seek to explore the differences in help-seeking of young adolescence into later adolescence and even young adulthood.

Limitations of this study include that the sample was predominantly from Victoria, Australia. At the time of data collection, Victoria was under a lockdown more severe than other states; therefore, this may have skewed the data, with more young people experiencing distress during this time.

## 5. Conclusions

Psychological distress is increasing among Australian youths [32], and COVID-19 has worsened how young people respond to adversity [13]. Young people were more likely to seek help from informal sources, such as friends and family. When experiencing suicidal ideation, young people reported the intention to seek help from a mental health professional; however, this was not the highest rated source of help, nor was it rated as extremely likely. Youth self-reported seeking help from a mental health professional but, they were extremely unlikely to do so. This is alarming given the increasing rate of death by suicide in the Australian youth population [61]. More needs to be done in the wake of COVID-19 to support youth mental health. Schools represent access to a large population of young people and could be an optimal environment to increase resilience and help-seeking to prevent the development of severe mental illness for Australian youths.

## Figures and Tables

**Figure 1 behavsci-13-00121-f001:**
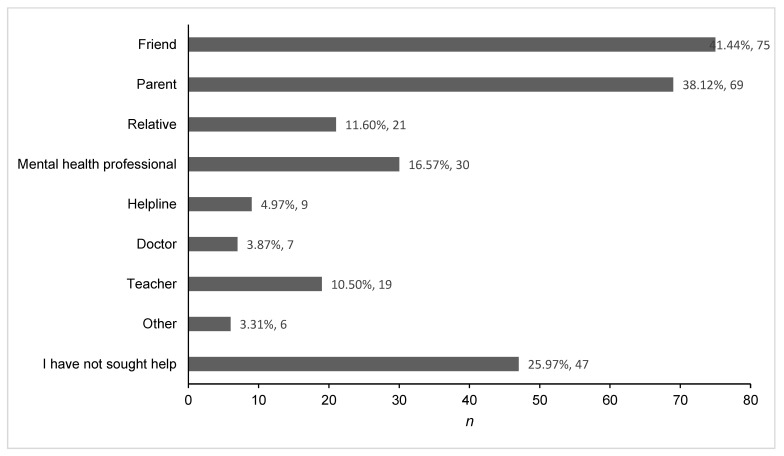
Sources of help sought from in the past two weeks.

**Figure 2 behavsci-13-00121-f002:**
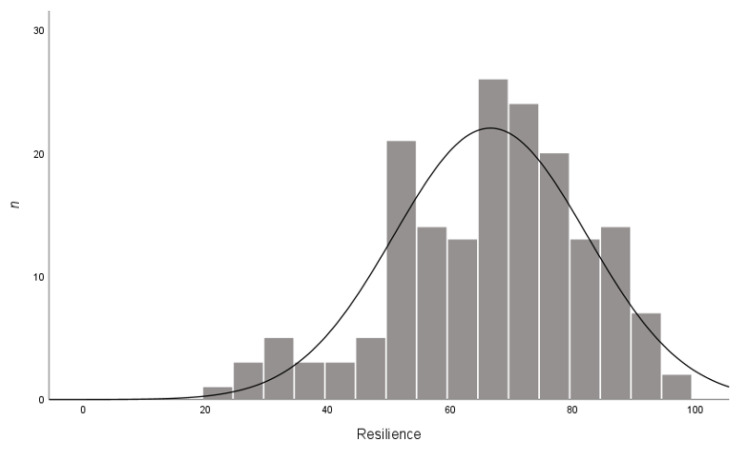
Resilience responses.

**Table 1 behavsci-13-00121-t001:** Help-seeking intentions.

		*n*	Mean	Standard Deviation
GHSQ Partner	Personal problem	179	4.38	1.99
	Suicide	180	4.21	2.12
GHSQ Friend	Personal problem	179	4.69	1.67
	Suicide	181	4.49	1.90
GHSQ Parent	Personal problem	178	4.85	2.18
	Suicide	181	4.72	2.29
GHSQ Relative	Personal problem	181	3.76	1.95
	Suicide	181	3.69	2.22
GHSQ Mental Health Professional	Personal problem	181	4.97	1.75
	Suicide	181	4.88	1.97
GHSQ Helpline	Personal problem	181	3.41	1.80
	Suicide	181	3.91	2.01
GHSQ Doctor	Personal problem	181	4.03	1.74
	Suicide	181	3.85	1.96
GHSQ Religious Leader	Personal problem	181	2.77	1.92
	Suicide	180	2.81	2.02
GHSQ Not seek help	Personal problem	181	2.88	1.94
	Suicide	179	2.72	2.02
GHSQ Other	Personal problem	99	2.39	1.97
	Suicide	95	2.25	1.95

**Table 2 behavsci-13-00121-t002:** Reported problems participants had sought help for in the past two weeks.

	Partner	Friend	Parent	Relative	Mental Health Professional	Helpline	Doctor	Teacher
Anxiety	2	12	8	1	5			4
Depression	2	11	7	1	9	1	2	2
Suicidal ideation	1	2			1	1		
Suicide attempt			1		1		1	
Self-harm					2			
Eating disorder					1			
Self-esteem		3						1
Sense of worthlessness	1	1			2			
Homelessness	1	1				1		1
General wellbeing	2	3	4					1
Parental separation	1	1	1	1				
Grief	1	1						
Changing identity	1	1	1					
Relationship problems	1	10	9	1	2			1
Moving overseas	1							
Bullying		1	4	1				2
Family problems		6	4		1	1		
Loneliness		3		1				
Behaviour		1						
Gender dysphoria		1			1			
Problems at school	1	7	7	1	1			1
Stress		10	7	2				3
Lockdown		1	3					1
Medication			1					
Future				1				
Trauma				1	4			

## Data Availability

Data will be uploaded onto Bridges (an online research repository) and available upon request.

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
