# Peer review of "Australian Youth Resilience and Help-Seeking during COVID-19: A Cross-Sectional Study"

_behavsci, 2023, doi:10.3390/bs13020121_

Round 1
Reviewer 1 Report
Thank you for submitting this article Australian Youth Resilience and Help Seeking during COVID-19: A Cross-Sectional StudyThe Covid-19 is a very interesting topic not only for the adolescent population in Australia, but for all adolescents worldwide. As you have seen Covid-19 has caused many mental disorders such as anxiety, stress and depression.
I would like to clarify some doubts that arise during the manuscript.
- line 55-57- You speak of protective factors although you do not indicate in the introduction which are the protective factors.
“ The protective factor model of resilience, guided by Resiliency Theory, posits that protective and risk factors interact. Therefore, by increasing protective factors this can reduce potential negative outcomes despite adverse life events ((Fergus & Zimmerman, 2005; Ledesma, 2014; Werner & Smith, 2001)
- Line 86- in your statement, “ Early intervention in youth mental health is associated with better outcomes into adulthood” , it is very general and does not indicate anything, please be more explicit.
- It relates in the results several variables, it would be important to see the relationship of the intention to seek help with the actual help-seeking by adolescents.
- The problems reported by the participants in Table 2, where did they get them from, are they open-ended responses to the questionnaire?
- Line 273- It is NOT clear what this sentence means, it would be interesting to rewrite it.
“Young people report that a major barrier to seeking help and using positive coping strategies to build resilience is low mental health literacy (Gulliver et al., 2010; Marinucci et al., 2022b)”
- The mean age of the participants was 13.76 years, it is known that the behavior of young people changes from childhood to adulthood. It is important to reflect this in the discussion since their inclusion criterion is 12 to 17 years.
I hope you will clarify your doubts so that your manuscript can be published.
Author Response
Thank you for the helpful suggestions.
We have responded by providing more clarity in the paper.
Please see the attachment.

Reviewer 2 Report
Dear Authors,
The theme of the article is recent and relevant, given the impact that the Covid-19 pandemic can have on the mental health of young people. The article is consistent and its sections are properly integrated.
In the introduction, the purpose and reason for the investigation are demonstrated. Statistical methods are appropriate and well-founded. The authors present the practical implications of their research.
However, authors should make the following changes:
- mention the psychometric characteristics of the Actual Help-Seeking Questionnaire , namely the internal consistency;
- refer to suggestions for future investigations based on the results of the research presented in the article.
Author Response
Thank you for your suggestions, we have provided them in the paper now.
Thank you also for the positive comments, we appreciate the need to have more research looking at the phenomena of the pandemics impact on this generation of adolescence.
Please see the attachment.
